# Replacing Missing Values in the Standard MISR Radiometric Camera-by-Camera Cloud Mask (RCCM) Data Product

Michel M. Verstraete[1], Linda A. Hunt[2], Hugo De Lemos[1], and Larry Di Girolamo[3]

[1]Global Change Institute (GCI), University of the Witwatersrand, Braamfontein, Republic of South Africa.
[2]Science Systems and Applications, Inc. (SSAI), Hampton, VA 23666-5845, USA.
[3]Department of Atmospheric Sciences, University of Illinois at Urbana-Champaign, Urbana, IL 61801, USA.

**Correspondence:** Michel M. Verstraete (Michel.Verstraete@wits.ac.za or MMVerstraete@gmail.com)

**Abstract.** The Multi-angle Imaging SpectroRadiometer (MISR) is one of the five instruments hosted on-board the NASA Terra platform, launched on 18 December 1999. This instrument has been operational since 24 February 2000 and is still acquiring Earth Observation data as of this writing. The primary missions of MISR are to document the state and properties of the atmosphere, and in particular the clouds and aerosols it contains, as well as the planetary surface, on the basis of 36 data channels collectively gathered by its nine cameras (pointing in different directions along the orbital track) in four spectral bands (blue, green, red and near-infrared). The Radiometric Camera-by-Camera Cloud Mask (RCCM) is derived from the calibrated measurements at the nominal top of the atmosphere, and is provided separately for each of the nine cameras. This RCCM data product is permanently archived at the NASA Atmospheric Science Data Center (ASDC) in Hampton, VA, USA and is openly accessible (Diner et al. (1999b) and https://doi.org/10.5067/Terra/MISR/MIRCCM_L2.004). For various technical reasons described in this paper, this RCCM product exhibits missing data, even though an estimate of the clear or cloudy status of the environment at each individual observed location can be deduced from the available measurements. The aims of this paper are (1) to describe how to replace over 99% of the missing values by estimates and (2) to briefly describe the software to replace missing RCCM values, which is openly available to the community from the GitHub web site https://github.com/mmverstraete or https://doi.org/10.5281/zenodo.3519901. Two additional sets of resources are also made available on the Research Data Repository of GFZ Data Services in conjunction with this paper: (A) The first set (Verstraete et al. (2020), https://doi.org/10.5880/fidgeo.2020.004) includes 3 items: (A1) a compressed archive `RCCM_Out.zip` containing all intermediary, final and ancillary outputs created while generating the Figures of this manuscript, (A2) a User Manual `RCCM_Out.pdf` describing how to install, uncompress and explore those files, and (A3) a separate input MISR data archive `RCCM_input_68050.zip` for Path 168, Orbit 68050. This latter archive is usable with (B) the second set (Verstraete and Vogt (2020), https://doi.org/10.5880/fidgeo.2020.008), which includes (B1) a stand-alone, self-contained, executable version of the RCCM correction codes `RCCM_Soft_Win.zip` using the IDL Virtual Machine technology that does not require a paid IDL license, as well as (B2) a User Manual `RCCM_Soft_Win.pdf` to explain how to install, uncompress and use this software.

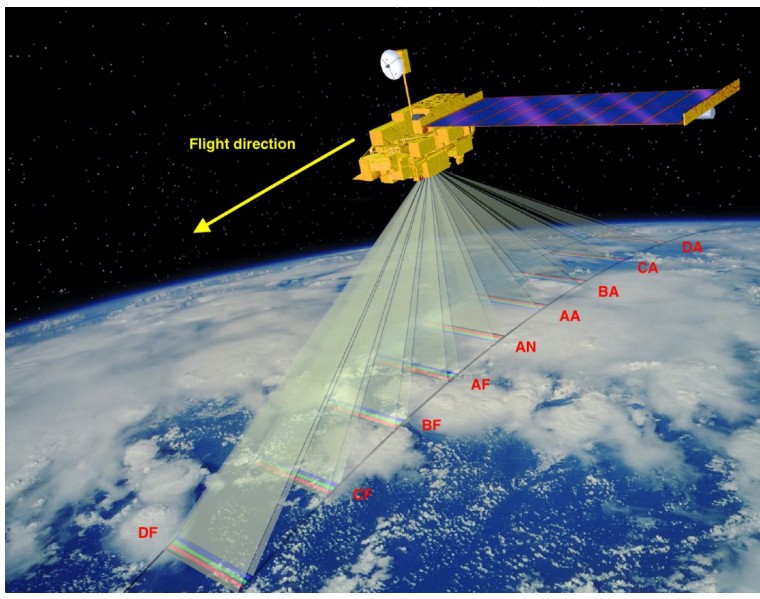

**Figure 1.** Artist rendition of MISR's observational protocol, with 9 cameras pointing in different directions along-track, each featuring 4 spectral bands. Image courtesy of Shigeru Suzuki and Eric M. De Jong, Solar System Visualization Project. Source: JPL image P-49081.

## 1 Introduction

The Multi-angle Imaging SpectroRadiometer (MISR) is a unique instrument that features nine "pushbroom" imaging cameras, each equipped to acquire data in four spectral bands (blue, green, red and near-infrared), thereby offering 36 data channels to
describe the properties of the Earth's environment (see Figure 1). These nine cameras, labeled DF, CF, BF, AF, AN, AA, BA, CA, and DA, are oriented at various bore angles along the orbital track of the Terra platform to acquire measurements at observers' zenith angles ranging from 70.3° forward to 70.6° aft. This instrument and its numerous applications have been amply described in the literature (Diner et al., 1989, 1998, 1999a, 2005); a complete list of all publications about the MISR instrument and data products is available at https://www-misr.jpl.nasa.gov/publications/peerReviewed/.

The spectral detectors and the optical lens systems of the various MISR cameras are designed to yield an effective spatial resolution across-track of 275 m in the eight off-nadir pointing cameras. For cost-related reasons, the nadir pointing camera AN is identical to the next two off-nadir pointing cameras AF and AA, which results in a slightly finer spatial resolution of 250 m. However, those data are re-sampled to 275 m as part of the Level 1 processing. Furthermore, to reduce the amount of data that need to be transmitted to the ground segment, raw data are averaged to a spatial resolution of 1100 m in the non-red data
channels of the off-nadir pointing cameras. Hence, in the default Global Mode of operation, the four spectral bands of the nadir pointing camera and the red spectral band in the eight off-nadir cameras are available at the native spatial resolution of 275

m, while all other data channels are only available at the reduced spatial resolution. MISR is occasionally operated in Local Mode, in which case all 36 data channels are available at the native spatial resolution of 275 m for a limited area along-track, but this type of acquisition will not be considered further in this paper as the cloud mask discussed in this paper is only derived in Global Mode.

The orbit of the Terra platform is carefully managed to maintain its altitude as well as the Equatorial Crossing Time (ECT) on its way from the North Pole to the South Pole. The original specification called for no more than 15 minutes deviation from the nominal 10:30 AM local time, but was tightened to within one minute after two years of operations. This platform completes sequences of 233 orbits in 16 days, after which it returns to observe the planet from the same relative position. In the MISR context, every orbit thus falls within one of these 233 fixed patterns, called "Paths". Each orbit is numbered

sequentially, starting with 1 at launch. The MISR instrument collects data only between the northern and southern terminators, while over-flying the sunlit side of the Earth, and the data acquired during that time are referred to as an "Orbit", even though it actually comprises only about half of the orbit. MISR became operational on 24 February 2000, while performing Orbit 995, and completed its 100,000th Orbit on 6 October 2018; it is still operational as of this writing, and therefore offers a unique opportunity to study environmental issues over a continuous span of 20+ years.

To facilitate the analysis of MISR data, the maximum possible daylit extent in each Orbit is segmented into 180 "Blocks" that have a fixed geographical location (as long as the orbit is properly maintained). Each Block is a rectangular area of 563.2 km across-track by 140.8 km along-track, which is large enough to contain the swath of the instrument. Successive Blocks of a given Orbit are staggered with respect to each other to account for the inclined orbit of the platform with respect to the equatorial plane. For any given Orbit, only about 142 Blocks contain actual data (Bull et al., 2011, p. 306).

Lastly, at Level 1B2, each and every individual MISR measurement (i.e., pixel value) is coded as a 16-bit unsigned integer, where the 14 most significant bits carry the scaled radiance information and the last 2 bits contain the Radiometric Data Quality Indicator (RDQI), an indicator of the quality of that particular measurement. This RDQI can take on the following binary values (Bull et al., 2011, p. 190):

- Binary `00` (integer `0`): the pixel value is "Within specifications".

• Binary `01` (integer `1`): the pixel value is deemed of "Reduced accuracy".

- Binary `10` (integer `2`): the pixel value is deemed "Not usable for science".

- Binary `11` (integer `3`): the pixel value is "Unusable for any purpose".

In this paper and the associated software, those values are labeled "good", "fair", "poor" and "bad", respectively.

## 2  Motivation

The standard MISR Radiometric Camera-by-Camera Cloud Mask (RCCM) data product (Diner et al. (1999b) and https://doi.org/10.5067/Terra/MISR/MIRCCM_L2.004) contains an 8-bit unsigned integer value for each pixel within each Block.

Different algorithms are applied over oceanic and terrestrial areas to determine the state of cloudiness at each geographical location, at spatial resolution of 1.1 km. The RCCM is generated for each of the 9 cameras using two different spectral bands, namely the red and near-infrared bands. All results discussed in this paper refer to the RCCM product Version F04_0025, which is the most current version available at the time of writing.

Over oceans, one of the tests used to generate the RCCM relies on the bidirectional reflectance factor (BRF) in the near-infrared band at 1.1 km resolution, while the other examines the standard deviation of the $4 \times 4$ array of the 275 m red band BRF within a 1.1 km squared area. The near-infrared band BRF and the red band standard deviation results for each pixel are each tested against three thresholds to classify a pixel as high-confidence cloudy, low-confidence cloudy, low-confidence clear, or high-confidence clear. The two tests may return different results, and the final cloud mask is determined from the logical

combination of the results of the two tests.

     Over land, the standard deviation test remains the same as over ocean, but the near-infrared BRF test is replaced with a vegetation index constructed from a combination of red and near-infrared BRFs. The logic for combining the two tests is different over land than it is over ocean, with a lower reliance on the standard deviation test over land. These algorithms are described in detail in the Level 1 Cloud Detection Algorithm Theoretical Basis Document (ATBD) (Diner et al., 1999b),

which can be downloaded from https://eospso.gsfc.nasa.gov/atbd-category/45. Updates to the RCCM are provided in Zhao and Di Girolamo (2004) and Yang et al. (2007). Validation is described within the GEWEX Cloud Assessment Report (Stubenrauch et al., 2012), which also shows that the RCCM compares favorably to many other satellite products when used to compute cloud fraction over snow- and ice-free surfaces.

     The RCCM values, which are unsigned 8-bit (or single byte) integers, carry the following meanings (Bull et al., 2011, p. 85):

• `0`: No retrieval.

     • `1`: Cloud with high confidence.

     • `2`: Cloud with low confidence.

     • `3`: Clear with low confidence.

     • `4`: Clear with high confidence.

• `255`: Fill value.

     This data product typically contains missing (no retrieval) data in the following four cases:

1. As part of the Level 1 processing, measurements are re-projected to a Path-specific Space Oblique Mercator (SOM) map projection, which minimizes distortions and scale errors (see, in particular, Appendix A of Bull et al., 2011, p. 299). This re-projection is implemented both on the World Geodetic System 1984 (WGS84) reference ellipsoid (for

oceanic applications and is referred to as Ellipsoid projected) and on a representation of the actual topography of the continents (referred to as Terrain projected). Over ocean, Terrain and Ellipsoid projected radiances are equivalent, so Terrain projected radiances are not reported over ocean. Within any camera, but most notably for those observing the

Earth at larger zenith angles, some ground areas may be obscured by the local terrain in the Terrain projected radiance (e.g., the surface area may lie behind hills or mountains, and may be unobservable from the point of view of those cameras—hence, no Terrain projected radiance are reported for those locations in those cameras).

2. The data files containing RCCM data cover geographical areas that are wider than the effective swath width of the MISR instrument. Hence, in any particular Block of data, the western and eastern sides of the rectangular Block are filled with a particular code indicating where no observations are available: these are referred to as *swath edge* pixels.

3. Occasionally, the on-board computer gets overwhelmed by the amount of data to be ingested. This may occur for various reasons, but happens most frequently when switching between Global and Local Mode, or conversely. Whenever this occurs, the computer quickly resets itself and restarts operations. These events are recorded as part of the initial raw data processing, and result in one or more *missing lines* of measurements. After re-projection onto the SOM map, these dropped lines appear as curved lines across the entire Block in the data.

4. The software that processes MISR data and generates the RCCM data product currently considers only L1B2 radiance data with an RDQI of `00`, and therefore ignores and dismisses all measurements considered of lower quality.

These various cases are exhibited in Figures 3 to 6 below. Figure 2 shows the geographical locations of those Blocks along the southeastern coast of South Africa and the Indian Ocean.

In the first two cases, no observations are made so these are truly missing values for which no estimate can be provided. In the third case, only missing values in the red or near-infrared spectral bands result in missing RCCM values. However, these missing lines affect different geographical locations in different spectral bands. This is because the red and NIR detectors lie next to each other on the camera focal planes, so that, at any given time, they actually observe locations separated by a few km on the ground. This shift in geolocation is taken care of during the L1B1 processing stage. The interesting outcome is that if an event results in the simultaneous dropping of a data line in both spectral bands, different locations are affected on the Earth's surface. In the fourth case, any radiance value in the red or NIR spectral band with an RDQI of `01` or higher would cause the RCCM to be missing over land, even though the cloudiness status of the area could be derived on the basis of available data.

Missing lines in L1B2 data have different implications for RCCM over water bodies and land masses. Over oceans, because the two tests use different spectral bands, one of them is often capable of estimating the cloudiness status of each pixel, as the geographical locations affected by missing lines varies with spectral band. In the case of land, the vegetation index test uses the red and the NIR band, so if either of those bands is unavailable, the vegetation test fails automatically. In this situation, the logic for combining the two tests over land will report Cloud with High Confidence only if the red band is available, contains 9 or more of the 16 275 m samples for computing the standard deviation of the red band reflectance, and passes the Cloud with High Confidence threshold; otherwise, it is flagged as "No Retrieval". Hence, there are many more cases of RCCM missing values over land.

Figure 3 shows a map of a particular Block of standard MISR RCCM data, where red pixels are deemed non-retrieved or "missing". The scattered isolated red dots within the mapped area correspond to locations that cannot be acquired by

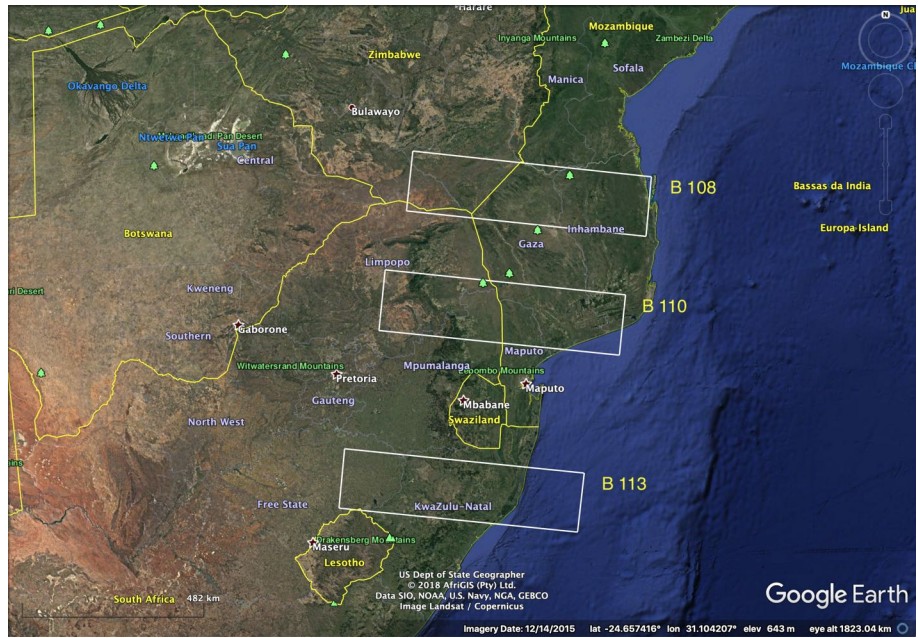

**Figure 2.** Google Earth Pro (Version 7.3.2.5776, 64-bit, March 5, 2019, 12:32:21 AM UTC) map of Southern Africa showing the geographical location of the three MISR Blocks mentioned in the text and in the figures of sections 2 to 4. NW corner: $-17.154$ (S) and 20.363 (E), SE corner: $-24.675$ (S) and 31.104 (E); eye altitude: 1823 km; data from SIO, NOAA, U.S. Navy, NGA, GEBCO and imagery from Landsat / Copernicus. Copyright 2018 AfriGIS (Pty) Ltd. URL: http://www.earth.google.com [accessed September 9, 2019].

that particular camera (the observation zenith angle for camera CA is 60.2°) due to obscuration by steep topography in the Drakensberg mountain range. The red areas on the western and eastern ends of the Block correspond to the swath edge areas: they fall outside the radiance swath of the MISR instrument for that camera and contain only fill values. The wide red curved line that runs across the entire area is an example of the effect of missing values in the red or near-infrared spectral bands of the MISR L1B2 data product (in this case, entirely over land).

Figure 4 shows a different example, in this case from another combination of Orbit and Block numbers. There are fewer scattered red dots because this map was derived for a less inclined camera (BF, with an observation zenith angle of 45.7°) observing a somewhat smoother terrain. There are again a couple of red curved lines running across the entire area, due to the missing red and near-infrared L1B2 radiance data. But there is also a linear feature aligned along-track near the western edge of the Block: in that case, the L1B2 radiance data are not actually missing, but associated with an RDQI of `01`.

Figure 5 exhibits a case where the numbers of missing lines in the red and NIR spectral bands of the L1B2 data are so large that they overlap on the Earth's surface. It can be seen that the thickness of the band of missing data in the RCCM product is larger over land than over the ocean. This is because one of the tests worked over water when only one spectral band was available, while neither test was operational over land. Note also the presence of another straight line of missing values along the western edge of the Block, due to L1B2 radiance values associated with an RDQI of `01`.

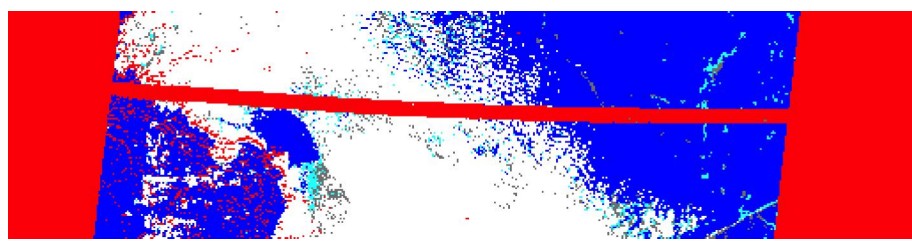

**Figure 3.** Map of the MISR RCCM product for Path 168, Orbit 68283, Block 110 and camera CA. All RCCM products are generated at the spatial resolution of 1100 m and provided as arrays of 512 by 128 pixels. The total size of the Block area is 563.2 km across-track by 140.8 km along-track, while the parallelogram-shaped ground area inside the Block is about 380 km across-track. Color coding: red: no retrieval (RCCM = 0) or fill value (RCCM = 255); white: cloud with high confidence (RCCM = 1); gray: cloud with low confidence (RCCM = 2); aqua: clear with low confidence (RCCM = 3); and blue: clear with high confidence (RCCM = 4).

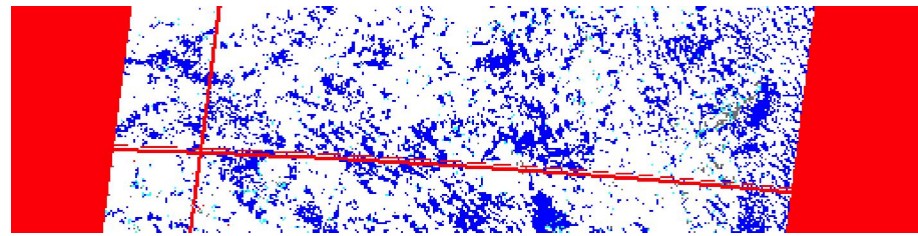

**Figure 4.** Map of the MISR RCCM for Path 168, Orbit 1412, Block 108 and camera BF. The linear dimensions of the mapped area and the color coding are identical to those indicated in Figure 3.

Figure 6 exhibits an extreme case of missing RCCM data. Here, most of the missing cloud mask values are due to L1B2 radiance data associated with an RDQI of 01 or higher, as can be seen from Figure 7, showing the RDQI values associated with the L1B2 radiance data in the red spectral band, one of the two used to determine the RCCM.

To be fair, it is important to note that missing L1B2 radiance data affect only a small proportion of all globally available data over any significant amount of time. However, these dropped lines tend to occur more systematically in connection with the switching between Global and Local Mode, as mentioned earlier, so they very much affect the usability of the products at those locations. Figure 8 shows how many pixel values went missing in the 9 cameras of the RCCM product for Path 168 and Block 110, which includes the Skukuza flux tower, a site initially monitored in Local Mode early in the mission in support of the Safari-2000 field campaign (see the special issue of the *Journal of Geophysical Research* dedicated to that campaign, openly available from https://agupubs.onlinelibrary.wiley.com/doi/toc/10.1002/(ISSN)2169-8996.SAF1), and systematically acquired since December 2009. A close inspection of the data reveals that straight lines of L1B2 radiance data with an RDQI value of 01 contaminated cameras DF and BF of just about every Orbit between the start of the mission and the end of March 2008. Between April 2008 and December 2009, four cameras were affected in this way: CF, BF, BA and DA. Very few missing RCCM values occurred during a short period of time (late December 2009 to early March 2010), even though this

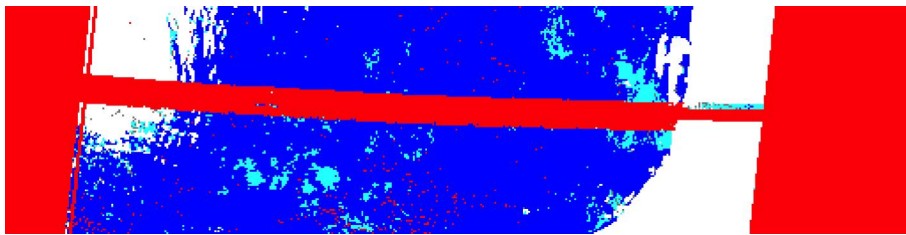

**Figure 5.** Map of the MISR RCCM for Path 168, Orbit 98340, Block 113 and camera BA. Note the variable thickness of the missing data band when crossing the shoreline between the continent and the Indian Ocean (see text for details). The linear dimensions of the mapped area and the color coding are identical to those indicated in Figure 3.

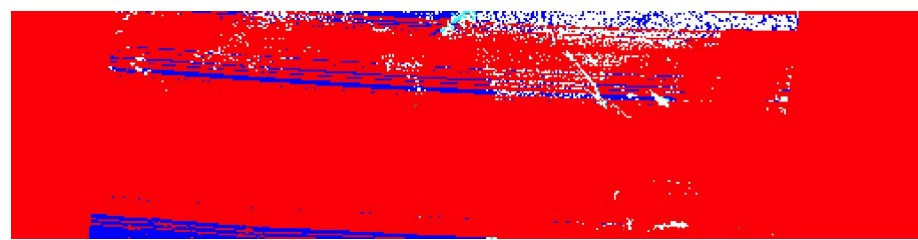

**Figure 6.** Map of the MISR RCCM for Path 168, Orbit 2111, Block 110 and camera BA. The linear dimensions of the mapped area and the color coding are identical to those indicated in Figure 3.

corresponded to the start of the systematic acquisition of Local Mode observations for that particular Path and Block. A highly variable number of missing values have been observed since March 2010.

Based on inspecting these and many other cases, we conclude that (i) some of the missing RCCM data actually result from missing L1B2 radiance data, in the form of dropped lines showing up as curved bands across the swath of the instrument, (ii) missing RCCM data also arise in conjunction with the presence of L1B2 radiance data associated with an RDQI of `01` or higher, which has historically contaminated a large number of Orbits, especially between the start of the mission and December 2009, and (iii) whenever measurements are missing, they may affect multiple cameras and spectral bands, but the geographical areas affected are different for each camera and band, due to the hardware configuration of the camera focal planes.

This suggests that analyzing the RCCM data in multiple cameras and spectral bands, and assuming some degree of spatial continuity in cloud fields, may be useful in establishing the clear or cloudy status of a geographical location where the standard RCCM data product available from the NASA Langley Atmospheric Science Data Center in Hampton, VA, USA, is missing.

The purpose of this paper is to describe a set of simple algorithms and associated software codes (publicly available from the GitHub web site https://github.com/mmverstraete or https://doi.org/10.5281/zenodo.3519901) that can be applied to the standard RCCM data product to generate an updated version of the product where most if not all of the missing values of type 3 and 4 mentioned at the start of this section have been replaced by estimates of the clear or cloudy status, based on the values actually available in other cameras and spectral bands. In turn, a more spatially complete RCCM product will be useful to users

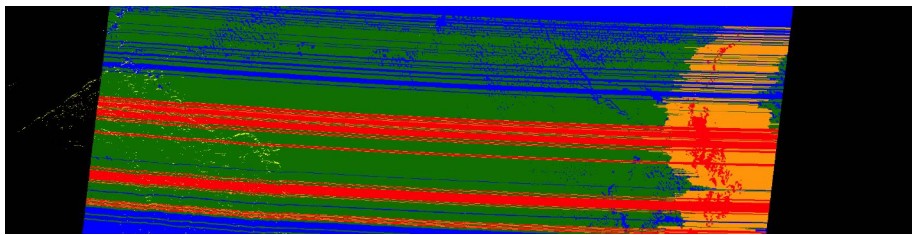

**Figure 7.** Map of the MISR L1B2 radiance quality data for Path 168, Orbit 2111, Block 110 and camera BA, in the red spectral band. This particular data channel is always available at the spatial resolution of 275 m. The linear dimensions of the mapped area are identical to those indicated in Figure 3. Color coding: black: edge pixels; yellow: area obscured by the local topography; blue: good data (RDQI = 00), green: fair data (RDQI = 01), gold: poor data (RDQI = 10), red: bad or missing data (RDQI = 11).

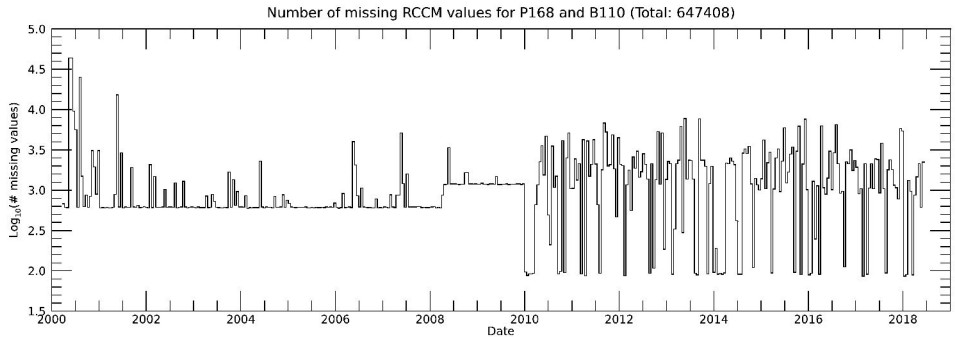

**Figure 8.** Plot of the number of missing RCCM values for Path 168 and Block 110, as a function of time (i.e., for all available Orbits) between the start of the mission and mid-2018 (about 650,000 out of a total of the order of 177 million values). This is a semi-logarithmic plot, so each vertical step of one unit represents an order of magnitude more missing points (see text for details).

of those data or derived products. In particular, updating this RCCM product is required for our next goal (to be addressed in a follow-up paper), which is to improve the spatial coverage of the L1B2 radiance product itself.

## 3 Processing

It is proposed to replace the missing values in the standard MISR RCCM data product by estimates of the clear or cloudy status of individual geographical locations in four successive steps, each implemented in a separate function written in the IDL language (IDL is a commercial software product distributed by Harris Geospatial Solutions, Inc.).

For each of the 9 MISR cameras, the following steps are implemented:

### 3.1 Reading data

The first step, implemented in IDL function `mk_rccm0.pro`, reads the standard MISR RCCM data product from the appropriate files for the specified Path, Orbit, Block and camera, downloaded from the NASA ASDC.

As indicated above, this original data set assigns a "No Retrieval" value to all RCCM pixels corresponding to MISR L1B2 radiance values associated with an RDQI other than `0`, independently of the origin of this data quality indicator value. This standard product therefore does not distinguish between unobservable locations (e.g., swath edge pixels or obscured by terrain) and genuinely missing observations.

### 3.2 Relabeling data

The second step, implemented in IDL function `mk_rccm1.pro`, accesses the standard MISR L1B2 scaled radiance and RDQI data products in the 4 spectral bands for the same Path, Orbit, Block and camera, in this case through pointers to data arrays that have been previously copied on the heap for efficiency reasons. The RCCM data array for the current camera is then updated as follows:

- If a pixel is deemed obscured (scaled radiance value with the RDQI attached of `65511`) in any one of the 4 spectral bands of the current camera, then that pixel value is set to `253` in the RCCM data array.

- If a pixel is classified as belonging to the swath edges of the Block (scaled radiance value with the RDQI attached of `65515`) in any one of the 4 spectral bands of the current camera, then that pixel value is set to `254` in the restored RCCM data array.

At this point, all remaining `0` pixel values in the RCCM array for the current camera indicate missing values that might be replaced by estimates.

### 3.3 Replacing missing values (same location, neighboring cameras)

In the third step, implemented in IDL function `mk_rccm2.pro`, the geographical locations of missing pixels in the current camera are retrieved. For each such missing RCCM value, the corresponding values at the very same geographical location (sample and line numbers within the Block) *in the preceding and following cameras*, in the natural sequence DF to DA mentioned earlier, are inspected. If these values are valid (not missing) and equal, i.e., if their values are in the range `[1, 4]` and identical in both cameras, then that same value is also assigned to the missing value in the current camera. This assignment assumes that if that particular location was regularly observed by the previous and the following cameras, and both of these cameras assign the same RCCM code defined in Section 2, then that location would likely have been assigned that same value if the measurement had not been missing from the current camera.

This part of the algorithm calls for two comments:

- Clouds always occur at some altitude above the ground. Hence, when different MISR cameras point to a particular location on the digital elevation model used in the terrain projection (as explained in Section 2), they may actually observe

different atmospheric volumes in altitude (parallax effect). This decision rule therefore assumes that the clear or cloudy areas observable by those neighboring cameras are actually continuous over the range of observation angles spanned by these three cameras. This rule is very effective and efficient in the case of large clear zones or consistent (overcast) cloud decks, but may lead to erroneous settings in the particular case where the cloud field is very discontinuous, i.e., composed of clouds (or clear areas) small enough to affect one camera but not its immediate neighbors. This limitation is mitigated by the fact that *two* neighboring cameras are inspected: if there is any discrepancy between their RCCM values, the replacement attempt is skipped at this stage.

- The two extreme cameras (DF and DA) do not have both a preceding or a following camera. In those cases, the two cameras following the DF camera (CF and BF) and the two cameras preceding DA (BA and CA) are inspected instead, using the same approach.

This step only addresses cases where there is consistency between the neighboring cameras regarding the clear or cloudy status of the locations for which there is missing information in the current camera. The remaining cases are more subtle, as they arise when neighboring cameras provide divergent assessments of the situation.

## 3.4 Replacing missing values (neighboring locations, same camera)

In this last step, implemented in IDL function `mk_rccm3.pro`, a decision concerning the probable clear or cloudy status of the remaining missing pixels of type 3 and 4 is based on the values of neighboring pixels *within the current camera*. This final assignment is implemented in 4 stages.

### 3.4.1 Stage A

The RCCM field for the current camera is first scanned to locate the remaining missing values. For each one of them, a small sub-window of 3 by 3 pixels is defined, centered on the missing pixel, and the valid RCCM values within that sub-window are inspected. If there are at least 4 valid (non-missing) values within that sub-window, *and* if all of them exhibit identical RCCM values, then that same value is assigned to the missing value. This rule is applied repeatedly, until no more missing values can be replaced using that decision rule. Hence, each time the Block is scanned to locate missing pixels meeting those requirements, values that have been reassigned previously can influence the assignment of neighboring missing RCCM values.

### 3.4.2 Stage B

If some missing values remain, then a new sub-window of 5 by 5 pixels is defined, again centered on each of the missing pixels. This time, a minimum of 12 valid values are required to make a decision, and they are allowed to differ from each other. The minimum (`min`), median (`med`) and maximum (`max`) RCCM values among those valid values are computed, and the following decision rules are applied:

- If $min = 1$, $max = 2$, and $med < 1.5$, then the missing value is set to $1$.

- If $\texttt{min} = 1$, $\texttt{max} = 2$, and $\texttt{med} \geq 1.5$, then the missing value is set to 2.

- If $\texttt{min} = 2$, $\texttt{max} = 3$, and $\texttt{med} < 2.5$, then the missing value is set to 2.

- If $\texttt{min} = 2$, $\texttt{max} = 3$, and $\texttt{med} \geq 2.5$, then the missing value is set to 3.

- If $\texttt{min} = 3$, $\texttt{max} = 4$, and $\texttt{med} < 3.5$, then the missing value is set to 3.

- If $\texttt{min} = 3$, $\texttt{max} = 4$, and $\texttt{med} \geq 3.5$, then the missing value is set to 4.

- If $\texttt{min} = 1$, $\texttt{max} = 3$, and $\texttt{med} < 1.5$, then the missing value is set to 1.

- If $\texttt{min} = 1$, $\texttt{max} = 3$, and $1.5 \leq \texttt{med} < 2.5$, then the missing value is set to 2.

- If $\texttt{min} = 1$, $\texttt{max} = 3$, and $\texttt{med} \geq 2.5$, then the missing value is set to 3.

- If $\texttt{min} = 2$, $\texttt{max} = 4$, and $\texttt{med} < 2.5$, then the missing value is set to 2.

- If $\texttt{min} = 2$, $\texttt{max} = 4$, and $2.5 \leq \texttt{med} < 3.5$, then the missing value is set to 3.

- If $\texttt{min} = 2$, $\texttt{max} = 4$, and $\texttt{med} \geq 3.5$, then the missing value is set to 4.

- If $\texttt{min} = 1$, $\texttt{max} = 4$, and $\texttt{med} < 1.5$, then the missing value is set to 1.

- If $\texttt{min} = 1$, $\texttt{max} = 4$, and $1.5 \leq \texttt{med} < 2.5$, then the missing value is set to 2.

- If $\texttt{min} = 1$, $\texttt{max} = 4$, and $2.5 \leq \texttt{med} < 3.5$, then the missing value is set to 3.

- If $\texttt{min} = 1$, $\texttt{max} = 4$, and $\texttt{med} \geq 3.5$, then the missing value is set to 4.

In effect, these rules are designed to assign an RCCM value consistent with the statistical distribution of values within the window under consideration, namely to assign the cloudiness level closest to the median of the neighboring values.

If the missing value is located near the northern or southern border of the Block, there may be fewer than the 9 or 25 values theoretically available in the 3 by 3 or 5 by 5 sub-windows. In those cases, the sub-windows are truncated and only valid values in those smaller sub-windows are considered. As before, this rule is repeatedly applied until no more missing values can be replaced.

### 3.4.3 Stage C

If the updated RCCM data product still contains missing values, the same algorithm as in stage B is applied, but this time requiring only 10 valid values to make a decision. As before, this rule is repeatedly applied until no more missing values can be replaced.

### 3.4.4 Stage D

Lastly, if there are some missing values left after stage C, the same algorithm is again applied, this time using the smaller 3 by 3 sub-window and requiring only 3 valid values in order to decide the clear or cloudy status of the missing value. As before, this rule is repeatedly applied until no more missing values can be replaced.

If there are still missing values after this last stage, the clear or cloudy status of the corresponding pixels is deemed impossible to determine. This actually occurs in only a handful of cases, when a very small area, corresponding to a single pixel in the reduced resolution data (hence of the order of 1.1 km by 1.1 km) is buried in mountainous terrain with fewer than 3 valid pixels in its immediate neighborhood.

### 3.5 Main function

The orderly application of those functions and the generation of outcomes is managed by the IDL function `fix_rccm.pro`, while the proper dimensioning of the sub-windows and the decision tree described above are implemented in another function called `repl_box.pro`.

    In summary, the RCCM array for each camera updated through this process takes on the following values: `0`: No retrieval, `1`: Cloud with high confidence, `2`: Cloud with low confidence, `3`: Clear with low confidence, `4`: Clear with high confidence,

`253`: Obscured pixel, `254`: Edge pixel, and `255`: Fill value.

### 4 Outcomes

The RCCM data arrays that have been updated in this way contain very few missing values, if any, and also correctly report on the location of obscured and edge pixels. This same function `fix_rccm.pro` also offers the opportunity to map the results obtained at the end of each step. For instance, the final updated RCCM data product for the same conditions as shown earlier

in Figure 3 is exhibited in Figure 9.

    The algorithm was able to correctly discriminate between the largely cloudy area in the middle of the scene and the mostly clear areas near both edges. It assigned most of the pixels within the main cloudy area to "cloudy with high confidence": this is, in part, the result of more inclined cameras observing more continuous cloud fields, and often constitutes the more conservative assessment (Zhao and Di Girolamo, 2004).

Figure 10 similarly exhibits the outcome of the process for the case shown earlier in Figure 4. In this case, the relative narrowness of the missing lines permitted the algorithm to restore most or all of the fine structure of the broken cloud field, without generating any noticeable artifact.

    The effectiveness of the process to replace missing RCCM values by estimates is really highlighted in the extreme case shown earlier in Figure 6, which has been restored as shown in the middle frame of Figure 11. To understand how that is

30 possible, the RCCM values for the previous and following cameras (AA and CA, respectively), are also exhibited. It can be seen that neither of those cameras is affected by missing values, so the algorithm described in subsection 3.3 is capable of

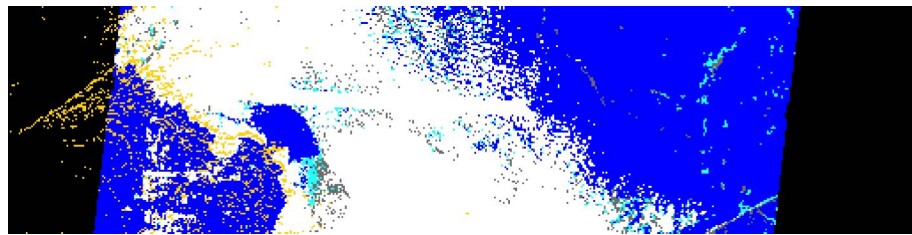

**Figure 9.** Map of the final updated (Stage D) MISR RCCM for Path 168, Orbit 68283, Block 110 and camera CA. Missing pixels have been replaced by estimates of the cloud or clear status of the observed areas, based on the cloudiness level of neighboring cameras and neighboring pixels within small sub-windows of the target camera, as described in the text. The enlargement and linear dimensions of the mapped area are identical to those indicated in Figure 3. Color coding: red: no retrieval or fill value; white: cloud with high confidence; gray: cloud with low confidence; aqua: clear with low confidence; blue: clear with high confidence; gold: pixels obscured by topography; and black: pixels in the edges of the instrument swath.

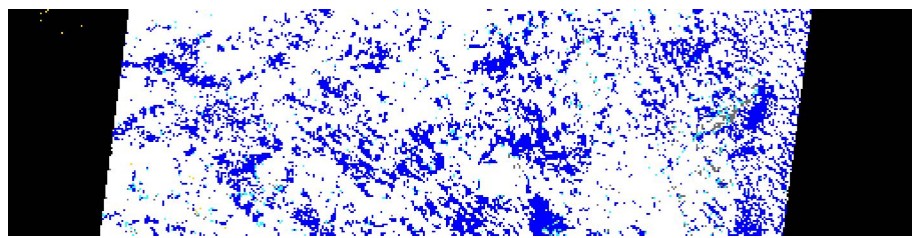

**Figure 10.** Map of the final updated (Stage D) MISR RCCM for Path 168, Orbit 1412, Block 108 and camera BF. Missing pixels have been replaced by estimates of the cloud or clear status of the observed areas, based on the cloudiness level of neighboring cameras and neighboring pixels within small sub-windows of the target camera, as described in the text. The enlargement and linear dimensions of the mapped area are identical to those indicated in Figure 3, while color coding is identical to that indicated in Figure 9.

restoring RCCM values for large consistently clear and cloudy areas. In fact, the outcome of that process (i.e., before applying the decision rules defined in subsection 3.4) can be seen in Figure 12.

Table 1 reports on the efficiency of the process to replace missing RCCM values in the various cases presented earlier in this paper. It can be seen that essentially all missing values are replaced by one of the four possible discrete estimates of cloudiness. The same rate of replacement has been found in all other cases inspected.

## 5 Algorithm performance

Last but not least, the accuracy of this algorithm needs to be evaluated. This is achieved by artificially inserting missing data in the RCCM product where none existed before, applying the replacement algorithm, and then comparing the replaced values to the original ones in a confusion matrix.

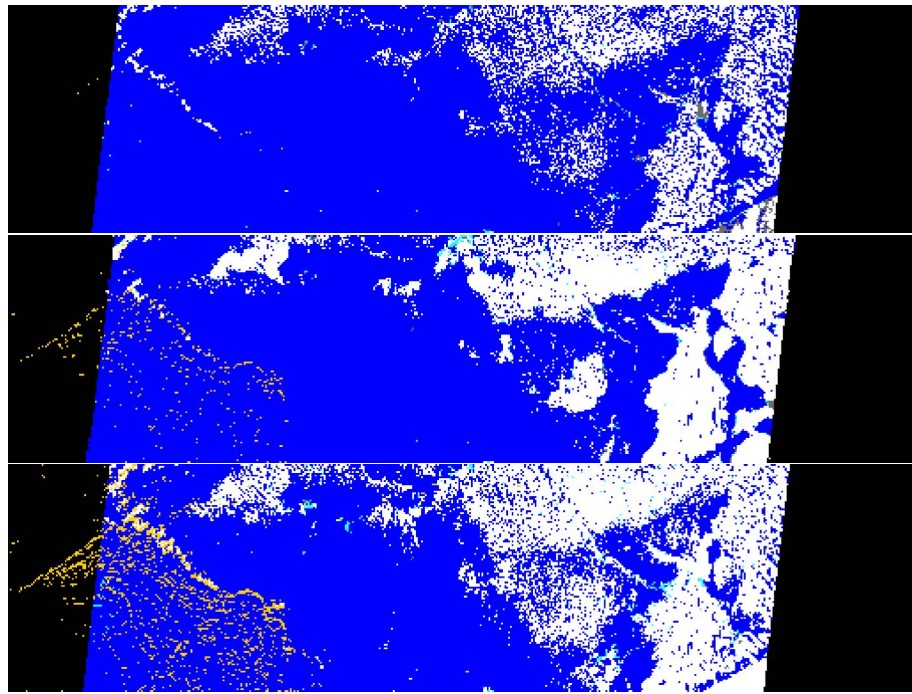

**Figure 11.** Maps of the MISR RCCM for Path 168, Orbit 2111, Block 110 and cameras AA (top frame), BA (middle frame) and CA (bottom frame). Restoring the clear and cloudy areas in camera BA (central frame), despite the very large number of missing values (as shown in Figure 6), is greatly helped by the absence of missing RCCM data in the AA (top) and CA (bottom) cameras. Note also the increased cloud cover, as well as the more prominent impact of the local topography, when the zenith angle of observation increases. Linear dimensions are as described in the legend of Figure 3, while color coding is identical to that indicated in Figure 9.

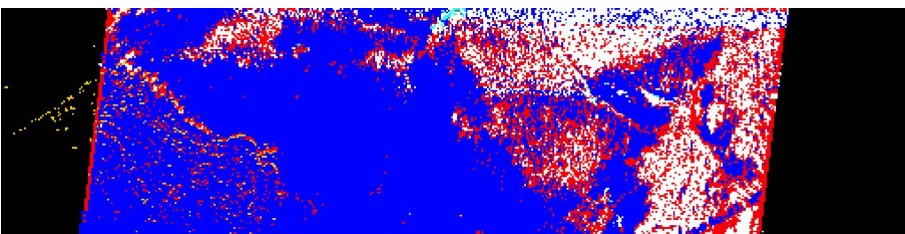

**Figure 12.** Map of the MISR RCCM for Path 168, Orbit 2111, Block 110 and camera BA. This map exhibits the intermediary product obtained at the end of the processing step involving values at the same location in the neighboring cameras (Subsection 3.3), and before investigating the values of immediate neighbors in the same camera (Subsection 3.4). Note also the presence of a straight line of missing values near the western edge of the Block, which is correctly replaced in the final corrected RCCM product shown in the middle frame of Figure 11. Linear dimensions are as described in the legend of Figure 3, while color coding is identical to that indicated in Figure 9.

The performance of the replacement algorithm will be documented by inspecting the results for various Blocks of Orbit 92981. The synoptic situation over a region further north on the same Path 168 as before, on that particular day (11 June 2017), is shown in Figure 13: nebulosity increases from almost clear to near overcast conditions between BLOCKS 101 and 107.

**Table 1.** Number of missing RCCM values remaining at the end of steps 1, 2 and 3 for the cases exhibited in this paper.

| Orbit | Block | n1 | n2 | n3 | Replacement rate |
|-------|-------|------|------|----|------------------|
| 01412 | 108 | 4947 | 2156 | 0 | 100.00% |
| 02111 | 110 | 43606 | 9555 | 1 | 99.99% |
| 68283 | 110 | 5337 | 1142 | 1 | 99.98% |
| 98340 | 113 | 7378 | 2184 | 0 | 100.00% |

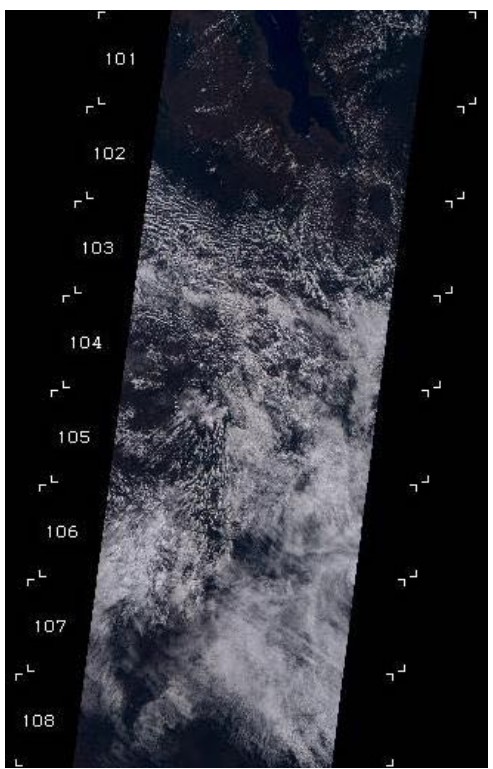

**Figure 13.** Fragment of the MISR Browse product for Path 168, Orbit 92981, Blocks 101 to 108, camera AN. The various Blocks in this scene, which covers central south Malawi and north west Mozambique, exhibit varying levels of cloudiness. See text for details.

Block 101 from that Orbit is mostly clear, with a few scattered clouds. The original RCCM data for cameras AF and CA did not contain any missing data, so some were artificially introduced on lines 60 to 64 of both cameras. The top panel of Figure 14 exhibits the original RCCM product, without any missing data. The middle panel shows where missing data were artificially introduced (lines 60 to 64), and the bottom panel displays the updated product, where missing data have been replaced by estimated values. Table 2 provides the confusion matrix for this case:

**Table 2.** Confusion matrix for P168, O092981, B101, camera AF.

| N = 1910 | Ori = 1 | Ori = 2 | Ori = 3 | Ori = 4 |
|---|---|---|---|---|
| New = 1 | 180 | 0 | 7 | 25 |
| New = 2 | 1 | 0 | 0 | 0 |
| New = 3 | 2 | 0 | 1 | 1 |
| New = 4 | 48 | 2 | 16 | 1627 |

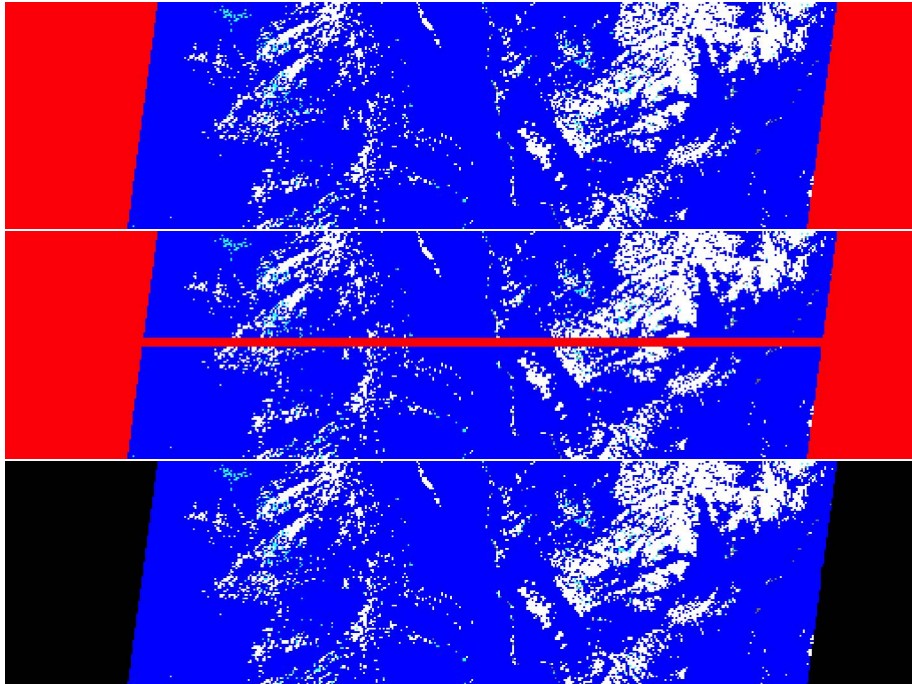

**Figure 14.** Maps of the RCCM for Path 168, Orbit 92981, Block 101, and camera AF. Linear dimensions are as described in the legend of Figure 3, while color coding is identical to that indicated in Figure 9. Top panel: map of the original RCCM product, which does not contain missing data; middle panel: map of the artificially modified RCCM product, where lines 60 to 64 were declared missing; bottom panel: map of the updated RCCM product, where the inserted missing data have been replaced by estimates.

Hence, 1808 values (94% of those missing) were correctly replaced, while 84 values (4% of those missing) were incorrectly classified, in the sense that the original classification was cloudy (or clear) while the replaced values were clear (or cloudy), respectively. This being said, only 1 pixel originally characterized as 'cloud with high confidence' was reassigned as 'cloud with low confidence', while 16 pixels originally characterized as 'clear with low confidence' were reassigned as 'clear with high confidence'. Similar results were obtained for camera CA: Table 3 shows the confusion matrix for that case.

**Table 3.** Confusion matrix for P168, O092981, B101, camera CA.

| N = 1879 | Ori = 1 | Ori = 2 | Ori = 3 | Ori = 4 |
|---|---|---|---|---|
| New = 1 | 237 | 3 | 2 | 47 |
| New = 2 | 1 | 0 | 0 | 2 |
| New = 3 | 5 | 1 | 1 | 1 |
| New = 4 | 108 | 4 | 3 | 1464 |

In that case, 1702 values (90% of those missing) were correctly replaced, while 169 values (8% of those missing) were incorrectly classified, in the sense that the original classification was cloudy (or clear) while the replaced values were clear (or cloudy), respectively. The results for this more inclined camera are marginally worse than for near-nadir pointing cameras, as expected, because the slanted view of small individual clouds may lead to mis-positioning the replacement values, especially if the clouds occur at high altitude.

A similar analysis conducted for Path 168, Orbit 92981 and Block 107, which covers an area that is largely overcast. In this case, cameras AA and CA did not feature any missing values in the original RCCM data. Some were artificially introduced from lines 30 to 34 in both cameras, and the results are exhibited in the confusion matrices of Tables 4 and 5, respectively.

**Table 4.** Confusion matrix for P168, O092981, B107, camera AA.

| N = 1911 | Ori = 1 | Ori = 2 | Ori = 3 | Ori = 4 |
|---|---|---|---|---|
| New = 1 | 1840 | 25 | 11 | 16 |
| New = 2 | 1 | 2 | 0 | 3 |
| New = 3 | 3 | 0 | 1 | 1 |
| New = 4 | 1 | 0 | 0 | 7 |

**Table 5.** Confusion matrix for P168, O092981, B107, camera CA.

| N = 1843 | Ori = 1 | Ori = 2 | Ori = 3 | Ori = 4 |
|---|---|---|---|---|
| New = 1 | 1835 | 7 | 1 | 0 |
| New = 2 | 0 | 0 | 0 | 0 |
| New = 3 | 0 | 0 | 0 | 0 |
| New = 4 | 0 | 0 | 0 | 0 |

Clearly, the replacement algorithm works very well for large overcast regions, as 96% and 99% of the missing values were correctly replaced in cameras AA and CA, respectively, and the performance of the replacement algorithm is not affected by the camera bore angle when the cloud deck is continuous.

The more challenging situation occurs when the scene includes a large number of isolated clouds, when these clouds occur at high altitude, when the camera affected by missing values is observing the Earth at large bore angles, and when there are many missing lines, especially if they are contiguous. The RCCM file for Camera DA of Block 102 in that sequence fits that description, as can be appreciated by inspecting Figure 13. It does contain native missing values, but more have been added in the northern part of the scene (lines 40 to 44). Figure 15 shows the original RCCM product (top panel), the modified version with additional missing data (middle panel), and the reconstituted version (bottom panel). The corresponding confusion matrix for this latter (worst) case is shown in Table 6.

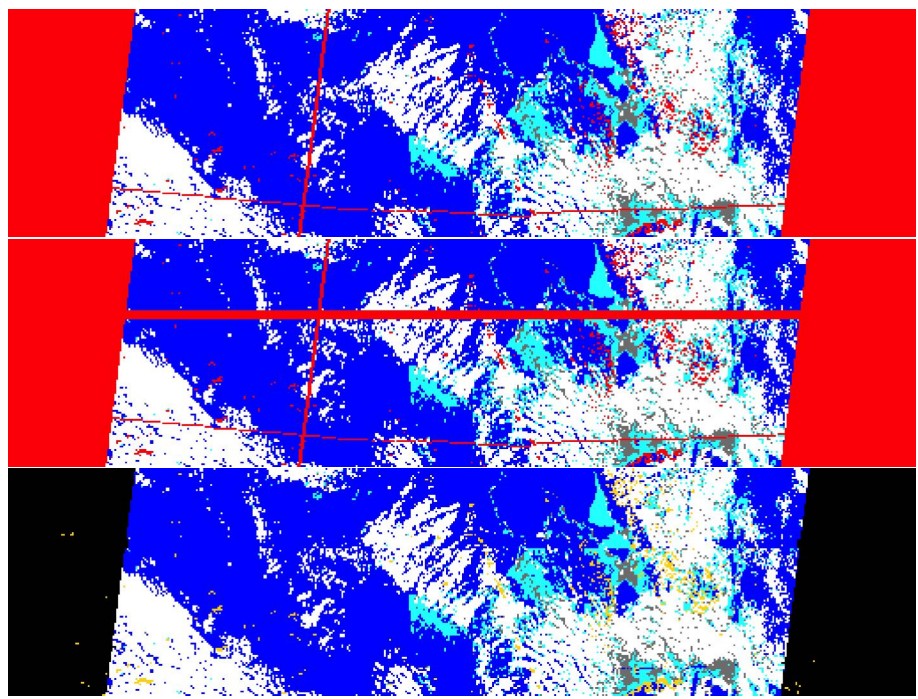

**Figure 15.** Maps of the RCCM for Path 168, Orbit 92981, Block 102, and camera DA. Linear dimensions are as described in the legend of Figure 3, while color coding is identical to that indicated in Figure 9. Top panel: map of the original RCCM product, which natively contain missing data; middle panel: map of the artificially modified RCCM product, where lines 40 to 44 were declared missing; bottom panel: map of the updated RCCM product, where all missing data have been replaced by estimates.

Some 1338 values (71% of those missing) were correctly assigned, while 343 values (or 18% of those missing) were incorrectly classified, in the sense that the original classification was cloudy (or clear) while the replaced values were clear (or cloudy), respectively. By the same token, 80% of the replaced values remained in their original category (clear or cloudy).

One last case will be presented to show that this replacement procedure may even yield more realistic results than the data generated by the original RCCM algorithm itself. Consider the case of P169, O078171, B102 and Camera DA: The original cloud mask is exhibited in Figure 16. The two types of missing lines (curved across-track and linear along-track) are present, but a quilting pattern of rectangular blocks of cloud values (1 and 2) is also quite prominent. In fact this quilting is apparent in

**Table 6.** Confusion matrix for P168, O092981, B102, camera DA.

| N = 1859 | Ori = 1 | Ori = 2 | Ori = 3 | Ori = 4 |
|---|---|---|---|---|
| New = 1 | 370 | 6 | 14 | 40 |
| New = 2 | 3 | 0 | 1 | 0 |
| New = 3 | 16 | 14 | 34 | 39 |
| New = 4 | 240 | 18 | 120 | 934 |

some of the other cameras too, though inspection of the L1B2 GRP data from which this product is derived shows that the state of the atmosphere on that day should be characterized as a clear day, with either humid conditions or perhaps a thin cirrus veil.

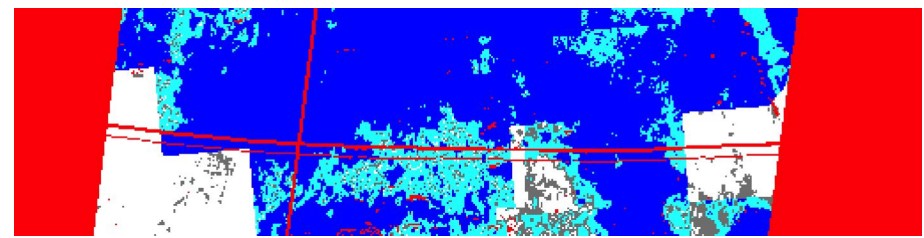

**Figure 16.** Map of the original RCCM for Path 169, Orbit 78171, Block 102, and Camera DA. Linear dimensions are as described in the legend of Figure 3, while color coding is identical to that indicated in Figure 9. This scene contains two curved lines of missing data across-track and one straight line of missing data along track. The cloud mask exhibits clear rectangular tiles which are the result of small changes in the dynamically determined thresholds used to classify the cloudiness level of the area.

Figure 17 shows the RGB maps of the top of atmosphere reflectance for the AN and DA cameras, respectively.

As an experiment, all RCCM data were artificially removed from Camera DA, and the cloud mask was regenerated on
the basis of the values in BA and CA, as described above. The result is shown in Figure 18. Some quilting pattern is still observable in the western part of the scene because the cloud mask in Camera CA also exhibits this pattern, but it has completely disappeared on the eastern side, and in any case the regenerated cloud mask for DA indicates a clear day throughout the scene, as can be visually observed in the RGB maps mentioned earlier.

This quilting pattern occasionally arises due to the nature of the original RCCM algorithm, which compares spectral index
values to dynamically computed thresholds, as explained at the start of Section 2. According to Catherine Moroney (personal communication, 5 January 2020), "this blockiness is due to discontinuities in the RCCM thresholds. Recall that the RCCM observables are compared against pre-determined threshold values in order to determine the actual cloud mask. These thresholds depend on the land-surface classification (one of 1580 values, as stored in the CSSC ancillary dataset at 1/6 degree resolution) and also the sun-view geometry. Sometimes, when you cross the border from one threshold bin (CSSC and sun-view angles)
to another, the threshold values themselves 'jump' so even though the observables themselves may be very similar in value, the thresholds that are used to determine cloud vs. clear are not."

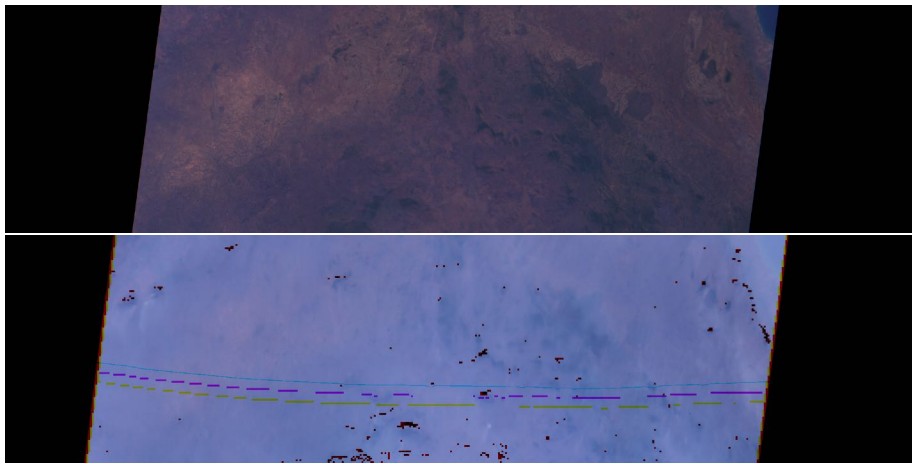

**Figure 17.** RGB maps of the original MISR L1B2 radiance product at the top of the atmosphere for Path 169, Orbit 78171, Block 102, and Cameras AN (top) and DA (bottom). Linear dimensions are as described in the legend of Figure 3. These maps show that the area was essentially clear throughout, possibly with a humid atmosphere or a thin cirrus veil. No rectangular clouds can be identified anywhere.

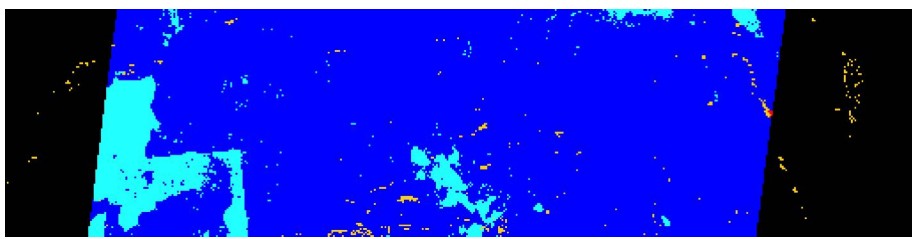

**Figure 18.** Map of the RCCM for Path 169, Orbit 78171, Block 102, and Camera DA, regenerated on the basis of the BA and CA values. Linear dimensions are as described in the legend of Figure 3. This derived product shows minimal quilting, and indicates that the area is clear throughout. See text for details.

It is of course not the purpose of this paper to address changes in the RCCM algorithms themselves, only to propose a solution to replace missing values. A future version of those algorithms may reduce or eliminate this quilting pattern in the cloud masks. In the meantime, if such quilting affects a particular camera more than its closest neighbors, that cloud mask could be reconstructed as explained above as a temporary fix.

5 **6   Conclusions**

The RCCM data product provides information on the spatial distribution of clouds in each of the 9 cameras of the MISR instrument. This data product contains missing values, which can be attributed to four different causes. Terrain obscured and swath edge pixels are actually not observable but are now properly identified in the corrected data product.

Missing values in the L1B2 radiance data product constitutes the third cause for missing RCCM values, as the latter is derived from an analysis of the former. This problem tends to happen randomly, but is much more frequent when the MISR instrument is switched from Global to Local Mode and conversely. Such changes in acquisition Mode affect only a small number of Paths and Orbits throughout the mission, but since some sites (such as the Skukuza site in the Kruger National Park) are systematically acquired at this full spatial resolution for scientific or calibration purposes, the areas surrounding those sites are then consistently contaminated by missing lines of data.

The systematic presence of L1B2 radiance data with an RDQI value higher than `00`, at least during the first 9 years of the MISR mission, constitutes the fourth process that leads to missing RCCM values. This condition may occur when the data acquired by a camera may be affected by sun glint, for instance, or whenever the result of resampling the raw data at Level 1 calls for a warning (Veljko Jovanovic, MISR Team at JPL, personal communication on 26 March 2019). More details can be found in the Algorithm Theoretical Basis document (Bruegge et al., 1999). In the experience of the authors, measurements associated with an RDQI of `01` are perfectly suitable for research applications, and their systematic exclusion in the generation of the RCCM is unwarranted. The procedures described above and the associated computer codes available from GitHub will effectively deal with those missing RCCM values, though it may be more consistent—in the context of a future reprocessing exercise—to allow the original RCCM algorithms to accept data with an RDQI of `01` in the first place.

The first two causal factors are entirely legitimate and missing data cannot be replaced by any reasonable value, however the algorithmic processing described in this paper results in the proper identification of those locations in the updated RCCM data product. The last two causal factors can be handled and replacement values can, in most cases, be proposed to fill in the missing RCCM values. The probable clear or cloudy state of those locations is derived from an analysis of valid (i.e., non-missing) RCCM values at the same location in the two neighboring cameras and/or at neighboring locations within the same camera.

Specific examples have been exhibited, including an extreme case where about 87% of the expected values in a RCCM data product are missing from a particular camera. In all cases, the RCCM values have been restored to visually reasonable and statistically conservative values. The only cases where the algorithms described in this paper cannot suggest a replacement value concern areas of size 1.1 km by 1.1 km (i.e., corresponding to a single pixel in the mapped product), isolated within mountainous areas, and with fewer than 3 neighboring pixels with valid values within the 3 by 3 or 5 by 5 sub-windows centered on the missing value itself.

The algorithm described in this paper effectively replaces 99% or more of the missing values, with an accuracy that ranges from 70% in the worst case to 99% in the best case. Synoptic situations with few isolated clouds or complete overcast are handled correctly, while scenes with many distinct clouds prove more challenging. As indicated earlier, all MISR L1B2 and RCCM data are freely and openly available from the NASA ASDC, and the IDL source codes used to generate the results described above are available under the MIT licence from the GitHub web site mentioned both above and below.

## 7   Code availability

This paper describes Version 2.1.5 of the RCCM IDL functions available in the following GitHub repository:

- https://github.com/mmverstraete/MISR_RCCM;
  DOI: https://doi.org/10.5281/zenodo.3519901

These functions, in turn, depend on more basic functions contained in separate repositories, listed here in order of increasing complexity:

- https://github.com/mmverstraete/Macros;
  DOI: https://doi.org/10.5281/zenodo.3519843

- https://github.com/mmverstraete/Utilities;
  DOI: https://doi.org/10.5281/zenodo.3519846

- https://github.com/mmverstraete/MISR_Tools;
  DOI: https://doi.org/10.5281/zenodo.3519861

- https://github.com/mmverstraete/MISR_L1B2;
  DOI: https://doi.org/10.5281/zenodo.3519989

Each function contained in those repositories contains abundant in-line documentation to describe every processing step. An HTML file documenting each function in a standardized format is also available in each repository. Functions in the more elaborate repositories may depend on routines defined in more basic repositories. It is thus recommended to download and install the contents of all those repositories to have access to the full set of tools. Please remember also that those tools are under active development and may be updated in time.

These IDL codes also depend on the following publicly available external tools:

- The MISR Toolkit Version 1.4.5 or later, available from
  https://github.com/nasa/MISR-Toolkit/releases

- An IDL routine called `match2.pro`, available from
  https://idlastro.gsfc.nasa.gov/ftp/pro/misc/match2.pro

- Three functions called `cgsetintersection`, `cgerrormsg` and `cgreverseindices`, available from
  http://www.idlcoyote.com/documents/programs.php

Lastly, for the benefit of users who may not have access to an IDL license, a license-free, self-contained, stand-alone, executable version of the software described above, using the IDL Virtual Machine technology, is available from the Research Data Repository of GFZ Data Services (see Verstraete and Vogt (2020); https://doi.org/10.5880/fidgeo.2020.008). A 'User Manual' describing the acquisition, installation and use of this software is also available from the same source.

## 8  Data availability

All MISR data, including the RCCM and the L1B2 radiance data products mentioned in this paper, were obtained from the NASA Langley Research Center Atmospheric Science Data Center. These data sets are openly available from the ASDC web site at: https://eosweb.larc.nasa.gov/project/misr/misr_table. The references for those data sets are Diner et al. (1999b) (https://doi.org/10.5067/Terra/MISR/MIRCCM_L2.004) for RCCM and Jovanovic et al. (1999) (https://doi.org/10.5067/Terra/MISR/MI1B2T_L1.003) for L1B2.

Two sets of resources are made available on the Research Data Repository of GFZ Data Services in conjunction with this paper: (A) The first set (Verstraete et al. (2020), https://doi.org/10.5880/fidgeo.2020.004) includes 3 items: (A1) a compressed archive `RCCM_Out.zip` containing all intermediary, final and ancillary outputs created while generating the Figures of this manuscript, (A2) a User Manual `RCCM_Out.pdf` describing how to install, uncompress and explore those files, and (A3) a separate input MISR data archive `RCCM_input_68050.zip` for Path 168, Orbit 68050. This latter archive is usable with (B) the second set (Verstraete and Vogt (2020), https://doi.org/10.5880/fidgeo.2020.008), which includes (B1) a stand-alone, self-contained, executable version of the RCCM correction codes `RCCM_Soft_Win.zip` using the IDL Virtual Machine technology that does not require a paid IDL license, as well as (B2) a User Manual `RCCM_Soft_Win.pdf` to explain how to install, uncompress and use this software.

*Author contributions.*  MMV: Co-development of the algorithm, software testing, senior author in charge of the initial and final drafting of the manuscript; LAH: co-development of the algorithm, manuscript updating; HDL: MISR data download, local processing, archiving of results for South Africa, testing the IDL VM package; LDG: description of standard RCCM algorithms, manuscript checking and updating.

*Competing interests.*  The co-authors of this manuscript declare that they have no competing or conflicting interests regarding the content of this paper.

*Disclaimer.*  The IDL source code software mentioned above (obtainable from the GitHub web site) is made available under the MIT license, which states, in part, that "The software is provided *as is*, without warranty of any kind, express or implied, including but not limited to the warranties of merchantability, fitness for a particular purpose and noninfringement. In no event shall the authors or copyright holders be liable for any claim, damages or other liability, whether in an action of contract, tort or otherwise, arising from, out of or in connection with the software or the use or other dealings in the software." Please read carefully and abide by the full license before downloading and using this software.

*Acknowledgements.* The first two authors (MMV and LAH) are greatly indebted to Prof. Bob Scholes (GCI at Wits University) for his unconditional scientific support for the MISR-HR project over the past decade, and to Prof. Barend Erasmus, Director of the University of the Witwatersrand's Global Change Institute (GCI) for sponsoring yearly visits to South Africa during the period 2016–2019. Hugo De Lemos is financially supported through a grant by Exxaro Limited to the Exxaro Chair in Global Change and Sustainability Research. The constructive comments of Catherine Moroney at NASA JPL, both public on the ESSD web site and private, are greatly appreciated. Dr. Peter Vogt, from the European Commission's Joint Research Centre (JRC) was instrumental in generating a self-contained, stand-alone, executable version of the RCCM correction software based on the IDL Virtual Machine technology. The computer used at GCI for processing MISR data was originally acquired through grant No. 97853 from the South African National Research Foundation (NRF). Last but not least, the first author (MMV) thanks Dr. David J. Diner of the NASA Jet Propulsion Laboratory (JPL), Principal Investigator of the MISR instrument, for co-opting him to the MISR Science Team some 25 years ago and for his encouragements ever since.

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
