# Peer review of "Replacing Missing Values in the Standard MISR Radiometric Camera-by-Camera Cloud Mask (RCCM) Data Product"

_Earth System Science Data, 2019_

## Short Comment (SC1) · 24 Jul 2019

This paper is well written and makes the valid point that while the number of "missing" pixels in the RCCM data may be small overall that they disproportionately affect areas close to local-mode sites which can be a real concern for people studying those particular regions of the earth.

Two comments:

(a) While the stated fill value of the RCCM is 255, this only occurs for blocks outside the first-block to end-block range. Blocks within the visible range will only use 0

(NoRetrieval) as a fill value even for the extreme edges of the block that are off the swath-edge. I could not tell from the paper if this affects the algorithm at all.

(b) I could not tell from the paper if the authors considered the affect of sunglint on their filling algorithm. The RCCM is calculated for all available radiances and even though the singlint mask is available in the same product it does not enter into the RCCM calculations so the affected areas will almost always be classified as Cloudy even if they are in fact Clear. The filling-in algorithm as described does not mention checking the glitter mask so it may be biasing the filled-in RCCM pixels towards the Cloudy side because it is accepting "false-cloud-caused-by-glint" data values at face value.

---

## Author Comment (AC1) · 7 Aug 2019

Dear Catherine,

Thanks a lot for your comment. Here are some further thoughts on these questions:

1. In our investigation, we found that ALMOST ALL MISR RCCM data sets (from any Path, Orbit and Block) contain some missing values. The numbers remain rather small, compared to the total number of observations actually available, but are not null. As an example, here is a plot of the time series of the number of missing RCCM values for Path 190 and Block 062 (an area covering the central Mediterranean sea,

between Tunisia and Italy, far from any Local Mode acquisition sites), from the start of the mission to early 2019. It can be seen that most data sets feature between 300 and 400 missing values (i.e., per Block) throughout the entire period.

2. As you correctly point out, the fill values 255B may only be used outside of the Block range for which there are actually usable data (e.g., in polar regions during the local winter), while a null code indicates an unobserved or a missing value within that Block range. For practical reasons, the software code that generates the maps shown in the paper represent both codes in red, so that it can work everywhere. This has no impact on the processing, which only attempts to replace null codes.

3. Regarding the effect of sun glint, the statement "it does not enter into the RCCM calculations so the affected areas will almost always be classified as Cloudy even if they are Clear." is incorrect: The RCCM thresholds are binned by sun-view geometry, and they have been tuned to deal with sun glint in those sun-view geometry bins (Zhao and Di Girolamo, 2004). That said, errors may still occur and may lean towards calling clear strong glint regions over water as cloudy, though not "almost always": this process concerns only about 15% for the glint regions of the AN camera (Zhao and Di Girolamo, 2006). Publications assessing this issue for the other cameras don't exist, but one of us (LDG), having looked at RCCM fields for almost 20 years, estimates this effect to be about the same in affected non-nadir cameras: around 15%.

Nevertheless, the proposed updating of the RCCM data product amounts to an interpolation scheme where we rely on existing and confirmed cloudiness estimates in some cameras and bands to infer the cloudiness levels in other cameras and bands. There may be a marginal bias towards increased cloudiness over water bodies, but this would be mitigated by the fact that the algorithm considers two neighboring cameras. Lastly, this possible side effect would of course not occur over continental regions.

Thanks again for stimulating this discussion.

References
Zhao, G. and Di Girolamo, L.: A cloud fraction versus view angle technique for automatic in-scene evaluation of the MISR cloud mask, Journal of Applied Meteorology, 43, 860–869, https://doi.org/10.1175/1520-0450(2004)043<0860:ACFVVA>2.0.CO;2, 2004.

Zhao, G. and Di Girolamo. L.: Cloud fraction errors for trade wind cumuli from EOS-Terra instruments, Geophysical Research Letters, 33, (20), L20802, https://doi.org/10.1029/2006GL027088, 2006.

[Figure]

[Figure]

Fig. 1.

---

## Referee Comment (RC1) · Anonymous Referee #1 · 2 Sep 2019

Review for "Replacing Missing Values in the Standard MISR Radiometric Camera-by-Camera Cloud Mask (RCCM) Data Product"

This paper is well-written and easy to understand. It describes a sequential approach to using multiple MISR cameras and a single MISR camera to replace missing data in the MISR RCCM product. The approach is implementable based on the methods presented in the paper but implementing it is unnecessary because all of the software is open to the public via GitHub. The paper and underlying research have benefits in terms of both accessing and improving MISR data products. All of my comments below are of a minor nature.

Minor Comments:

1) On Line 11 of Page 8 the words "might prove useful" are found and just below on Line 18 the words "may be useful" occur. This is a weak motivation for the study. Reading between the lines, it would seem that the local mode site over Sukuza absolutely required development of this software in order to make the RCCM data useful at this location. Is this true? If so, motivate the study by stating that missing data must ("must" is strong; "might" is weak) be replaced to execute some studies. This software was developed to this end and its public availability will facilitate future studies using MISR data. If the statement above is not true, at least for some demonstrated studies, motivation for the algorithm remains weak.

2) On Page 14 the words "successfully" (Line 6) and "rate of success" (Line 7) occurred. Missing values have indeed been replaced by estimates but to attach the words "success" to them one must demonstrate that the estimates are correct. This leads to the question as to how to assess the "correct replacement percentage" as opposed to the "replacement percentage" produced by the algorithm. As the manuscript now stands, this point is not addressed. The "correct replacement percentage" will certainly depend on the cloud type. How about using many, many different cloud types over Sukuza and/or neighboring regions for which there are no missing data and then removing lines and pixels in a way that captures the statistics of the missing data at Sukuza and for a variety of cloud types. Then, one would be able to assess algorithm accuracy by replacing the missing data and then comparing it to its truth. This does not seem like a hard exercise to do and it, or something like it, would demonstrate the success rate of the replacements, thereby providing confidence in the algorithm. It would seem that a broken cloud field with a lot of edge pixels would lead to the worst performance. So perhaps limiting assessment to a single cloud field of this type would demonstrate algorithm accuracy in a worst case scenario.

Details:

[Figure]

0) The marked up manuscript returned to the authors contains suggestions and questions on really minor details that the authors may want to consider.

1) Lines 12-13, Abstract: Maybe "for replacing missing values in the MISR RCCM data products" would be better than "to process MISR RCCM data products".

2) Page 5, Line 13: "RCCM to be missing over land". Is the "over land" necessary in this statement? Why not ocean too?

3) Page 5, Second Sentence of Figure 2 caption: Is this sentence necessary?

4) Much of the discussion on Page 6 regarding the scenes would benefit from having a figure that shows where the blocks are located relative to underlying topography and water.

5) Page 8, Line 17: The reference to "type 3 and 4" is not perfectly clear because if falls under a paragraph on the same page with an itemized list with 3 items. Perhaps back on Page 4, Line 17, the words "four cases" could be replaced by "four types" and then the list of types could be labeled Type 1, Type 2, Type 3, Type 4. In this way use of the word type would be tied closely to this list.

6) Page 9, Lines 9-10: This sentence needs to be rewritten. How about "As indicated above, this MISR RCCM data product assigns a "No Retrieval" 0B value to all RCCM pixels corresponding to MISR L1B2 radiance values associated with an RDQI other than 00, independently of the value of the data quality indicator." Note that as the text now stands a value of 0B is associated with the RDQI but this is inconsistent with its values listed on Lines 15-18 on Page 3. This mix-up occurs in a number of places. I caught a few places but the authors may want to make sure that mix-ups are eliminated.

7) Page 13, Line 11: "Figure 4" is not correct in this last line on Page 13. It should be "Figure 5".

8) Page 14, Figure 10 caption: "Figure 4" should be replaced by "Figure 5" in the caption.

[Figure]

9) Page 15, Line 6: "0B" should be "00" for consistency with earlier definitions.

10) Page 15, Line 11: "1B" should be "01" for consistency with earlier definitions. Same issue on Line 14 of Page 5:

11) Page 15, Line 15: "The first two processes" might be better as "The first two types".

12) Page 16, Line 16: "HR" has not been defined previously.

Please also note the supplement to this comment:
https://www.earth-syst-sci-data-discuss.net/essd-2019-77/essd-2019-77-RC1-supplement.pdf

**Supplement:**

[revised manuscript text omitted]

---

## Author Comment (AC2) · 25 Sep 2019

Dear Anonymous Reviewer #1,

Thanks a lot for your constructive comments. We have addressed those as indicated in the following sections, and a new version of the manuscript is attached below. Best regards, Michel Verstraete, on behalf of all co-authors.

A. Minor comments

1) The writing on page 8 reflected our state of mind as we inspected the data, unveiled these correlations and hypothesized the proposed solution. It is true that the latter does

turn out to be very useful–in retrospect.

On the other hand, correcting the Local Mode data was not our primary motivation, only a side issue. Having access to a RCCM product as complete as possible is actually necessary to fill out the missing values in the L1B2 radiance product itself, a more complex topic that will be the subject of a follow-up paper.

To address both of those points, we have implemented the following changes:

* p. 8, l. 11: Replaced "might prove useful" by "may be useful".

* p. 8, l. 18-19: Replaced "In turn, a more spatially complete RCCM product may be useful to users of those data or derived products." by "In turn, a more spatially complete RCCM product will be useful to users of those data or derived products. In particular, updating this RCCM product is required for our next goal (to be addressed in a follow-up paper), which is to improve the spatial coverage of the L1B2 radiance product itself."

2) This second point is well taken. We have updated the software to allow the artificial insertion of missing values in the RCCM data and to document the performance of the replacement algorithm with the help of a confusion matrix. These materials, which make up an entirely new section (5) of the paper, occupy 4 additional pages and include 3 new Figures as well as 5 new Tables. The concluding section of the paper has also been modified accordingly. Note that the software has been updated to implement those evaluations, and that a new version will be uploaded on the open source GitHub web site, once this paper appears in final form, to ensure that the code and the publication are in sync.

B. Editorial remarks (on the marked up manuscript)

* p. 1, l. 5: Replaced "on the basis of 36 data channels gathered by each of its nine cameras" by "on the basis of 36 data channels collectively gathered by its nine cameras".

* p. 1, l. 9-10: Indicating a percentage (e.g., 0.1%), or even a range of percentages (e.g., 0.1 to 80%), of missing RCCM values in a short abstract is prone to misunderstanding because these values are so variable in space and time. Even an abbreviated outline of those statistics would unreasonably lengthen this paragraph, as most missing values appear over land (see below). We have opted to keep the original wording.

* p. 1, l. 12: Replaced "describe how to replace most missing values" by "describe how to replace over 99% of the missing values".

* p. 1, l. 13: Replaced "to process MISR RCCM data products" by "to replace missing RCCM values".

* p. 2, l. 5: Replaced the semi-colon by a comma.

* p. 2, l. 7: The expression "ground segment" designates the infrastructure that receives and processes the data from the satellite. This is the proper technical term.

* p. 2, l. 8-10: Replaced "Hence, the four spectral bands of the nadir pointing camera, and the red spectral band in the eight off-nadir cameras, are available at the native spatial resolution of 275 m, while all other data channels are only available at the reduced spatial resolution, in the default Global Mode of operation." by "Hence, in the default Global Mode of operation, the four spectral bands of the nadir pointing camera and the red spectral band in the eight off-nadir cameras are available at the native spatial resolution of 275 m, while all other data channels are only available at the reduced spatial resolution."

* p. 2, l. 10: Replaced "MISR can occasionally be operated in Local Mode" by "MISR is occasionally operated in Local Mode".

* p. 3, l. 4: Replaced "and has already completed over 100,000 Orbits" by "and completed its 100,000th Orbit on 6 October 2018".

* p. 3, l. 5: Replaced "range" by "span".

* p. 3, l. 6: Replaced "in" by "into".

* p. 3, l. 10: Replaced "Equatorial" by "equatorial".

* p. 3, l. 12: Deleted comma.

* p. 3, l. 22: Replaced "assigns" by "contains" and replace "to" by "for".

* p. 3, l. 29: Replaced "near-infrared BRF" by "near-infrared band BRF".

* p. 4, l. 6: Moved the opening parentheses in front of the dates rather than the names.

* p. 4, l. 27: The expression "The data files containing RCCM data" refers to the standard data files distributed by ASDC. These include data for a complete Orbit, i.e., from the northern to the southern terminator, which contain about 142 Blocks with effectively usable data, as explained on p. 3, l. 10. Their geographical coverage is wider than the swath width of the instrument, and this feature is independent from the notion of Blocks. The subsequent sentence only mentions "in any particular Block of data" because the updating software process works on individual Blocks, as is evident from all examples shown. No action taken.

* p. 4, l. 29: Replaced "these are referred as the swath edge pixels" by "these are referred to as swath edge pixels".

* p. 5, l. 4: Replaced "The software processor that interprets" by "The software that processes".

* p. 5, l. 6: Inserted a sentence to introduce the new Figure 2, showing the topography of Southern Africa and the location of the MISR Blocks used in subsequent Figures.

* p. 5, l. 7: Replaced the comma by a full stop.

* p. 5, l. 13: The question "What about ocean?" appears to have been written upon first reading, as this point is addressed in the very next paragraph. No action taken.

* p. 5, l. 14: Replaced "since" by "because".

* p. 5, l. 15: Replaced "because" by "as".

* p. 5, l. 16: Deleted "each".

* p. 5, l. 19: Added a space between "275" and "m" and replaced "red-band" by "red band".

* p. 5, l. 20: Added a comma after "otherwise".

* p. 5, l. 21-22: Replaced "These various cases are exhibited in the following three figures for the southeastern coast of South Africa and the Indian Ocean." by "These various cases are exhibited in Figures 3 to 6 below. Figure 2 shows the geographical locations of those Blocks along the southeastern coast of South Africa and the Indian Ocean."

* p. 5, legend of Figure 2: Deleted the sentence "These maps have been enlarged (4x in each direction) by duplication for viewing convenience and to facilitate comparisons with other maps."

* p. 6, top of the page: Added a new Figure 2, extracted from Google Earth, with the following legend: "Google Earth Pro (Version 7.3.2.5776, 64-bit, March 5, 2019, 12:32:21 AM UTC) map of Southern Africa showing the geographical location of the three MISR Blocks mentioned in the text and in the figures of sections 2 to 4. NW corner: -17.154 (S) and 20.363 (E), SE corner: -24.675 (S) and 31.104 (E); eye altitude: 1823 km; data from SIO, NOAA, U.S. Navy, NGA, GEBCO and imagery from Landsat / Copernicus. Copyright 2018 AfriGIS (Pty) Ltd. URL: http://www.earth.google.com [accessed September 9, 2019]."

* p. 6, legend of old Figure 4: Replaced "See" by "see".

* p. 6, l. 15: Replaced "could work over" by "worked over".

* p. 7, l. 1: Replaced "tests were" by "test was".

* p. 7, l. 3: Replaced the colon by a full stop.
* p. 8, l. 1: Replaced "occur" by "occurred" and replaced "corresponds" by "corresponded".

* p. 8, old Figure 7: Adding the total number of values acquired would require significant modifications to the codes, as the plotting routine does not currently keep track of that total. Instead, the total number of theoretically observable values during that period is now mentioned in the text itself.

* p. 8, legend of old Figure 7: Replaced "See" by "see".

* p. 8, l. 16: Replace "that" by "the" twice.

* p. 8, l. 17: Replace "type 3 and 4 above" by "type 3 and 4 mentioned at the start of this section" and relabel the three points in the paragraph immediately above as (i) to (iii) to prevent any possible confusion.

* p. 9, l. 4: Add a full stop.

* p. 9, l. 9 and 10: The notations and references to binary and byte (8-bit unsigned integer) values have been systematically updated throughout the manuscript, as described in section C below.

* p. 9, l. 10: Replace "independently from the reason for this latter quality indicator:" by "independently of the origin of this data quality indicator value." This version differs from that proposed by the reviewer because the key point is the reason (cause) for the RDQI, rather than its value.

* p. 10, l. 22: The "type 3 and 4" should be clear, after the changes made on p. 8, l. 17.

* p. 13, legend of old Figure 8: Replace "The remarks and the linear dimensions" by "The enlargement and linear dimensions".

* p. 13, l. 3: Deleted "It can be seen that".

* p. 13, legend of Figure 9: Replace "The remarks and the linear dimensions" by "The enlargement and linear dimensions".

* p. 13, l. 11: Replaced (old) "Figure 4" by (old) "Figure 5", which is now Figure 6.

* p. 14, l. 4: Deleted the comma.

* p. 14, legend of Figure 10: Replaced (old) "Figure 4" by (old) "Figure 5", which is now Figure 6.

* p. 14, l. 6: Deleted "successfully".

* p. 14, l. 7: Replaced "success" by "replacement".

* p. 14, l. 8: Added a new section on the evaluation of the replacement process.

* p. 15, legend of Table 1: Deleted "elsewhere".

* p. 15, line 1 of Table 1: Replaced "Success" by "Replacement".

* p. 15, l. 6, 11, 14: See section C below.

* p. 15, l. 15: Replaced "processes" by "causal factors".

* p. 16, l. 1: Replaced "processes" by "causal factors".

* p. 16, l. 1: Kept the original text, which is more appropriate than the suggested replacement.

* p. 16, l. 3: Deleted the comma.

* p. 16, l. 16: Deleted "and MISR-HR products" since those have not been introduced in this paper.

C. Changes in the notations and labels

To avoid any confusion, all references to the RDQI are now typeset as 2-digit binary numbers while all references to RCCM values are typeset as integers (without the 'B')

that designated 8-bit unsigned integers.

* p. 4, l. 9: Replaced "The RCCM values carry the following meanings" by "The RCCM values, which are unsigned 8-bit (or single byte) integers, carry the following meanings".

* p. 4, l. 10-15: Deleted 'B'

* p. 4, l. 16: Deleted this line.

* p. 5, l. 5: Replaced "0B" by "00".

* p. 5, l. 12: Replaced "1B" by "01".

* p. 5, legend of (old) Figure 2: Removed the "B" in the description of the color coding scheme.

* p. 6, l. 12: Replaced "1B" by "01".

* p. 7, l. 2: Replaced "1B" by "01".

* p. 7, l. 4: Replaced "1B" by "01".

* p. 7, legend of (old) Figure 6: Replaced the integer values of the RDQI by their binary representations, for consistency.

* p. 7, l. 14: Replaced "1B" by "01".

* p. 8, l. 6: Replaced "1B" by "01".

* p. 9, l. 10, 19, 21 and 23: Deleted "B".

* p. 9, l. 29-30: Replaced [1B, 4B] by [1, 4].

* p. 11, l. 6 to 21: Deleted all "B" characters.

* p. 12, l. 18 to 25: Deleted "B".

* p. 15, l. 6: Replaced "0B" by "00".

* p. 15, l. 11: Replaced "1B" by "01".

* p. 15, l. 14: Replaced "1B" by "01".

Please also note the supplement to this comment:
https://www.earth-syst-sci-data-discuss.net/essd-2019-77/essd-2019-77-AC2-
supplement.pdf
* * *
[Figure]

**Supplement:**

[revised manuscript text omitted]

---

## Referee Comment (RC2) · Anonymous Referee #1 · 30 Sep 2019

Building into the software an ability for its users to assess replacement performance is a nice touch. This will enable users to assess replacement performance on scenes of interest to themselves. This capability more than takes care of concerns with the initial content of the paper on this point.

---

## Author Comment (AC3) · 30 Sep 2019

Dear Anonymous Reviewer #2,

Thank you for your kind comment.

Best regards, Michel Verstraete, on behalf of all co-authors.

---

## Editor Comment (EC1) · Birgit Heim (Editor) · 16 Jan 2020

Dear Authors and Colleagues

thanks for the authors for the replies to the comments and reviews of your paper. The review and comments on your manuscript are positive. The manuscript is well-written presenting a useful and needed approach and published IDL code for a wider scientific user community of MISR satellite data. Of particular note is the good and detailed description of the methodology and data and higher level product characteristics.

ESSD policy requires easily accessible non-proprietary databases, data products, data

processing codes and other licence-free software tools necessary to process and use the data. For openness, replicability, and low entry threshold we ask the authors, in addition, to provide and publish a stand-alone program that is usable with the licence-free IDL Virtual Machine. We also recommend to publish a set of MISR test data to run with the proprietary version of IDL and the licence-free version within the IDL Virtual Machine environment.

Your ESSD publication has the large potential to trigger the interest, use and processing of MISR data also in the non-MISR user community. To enable easy use we recommend a code use documentation.

I look forward to your final manuscript, Best wishes, Birgit Heim

———————————————

---

## Author Comment (AC4) · 31 Jan 2020

Dear Birgit,

Thanks for your inputs on this manuscript.

The comments of the Reviewers have been taken into account as follows:

1. The question raised by Catherine Moroney (SC1) during the public discussion period was answered directly on the ESSD web site (AC1).

2. The comments made by the first Reviewer (RC1) were addressed in our formal responses also already available from the ESSD web site (AC2 and AC3).

[Figure]

3. Your remarks (EC1) have been taken into account by generating the following additional resources:

- A stand-alone, self-contained, executable version of the software to process MISR RCCM data and replace missing values, using the IDL Virtual Machine technology.

- A User Manual (in PDF format) explaining how to acquire, install and use this software package on a PC running under Microsoft Windows operating system.

- A ZIP archive of MISR L1B2 GRP and RCCM files for Path 168 and Orbit 68050 to serve as an example of input data: this allows users to explore how this software works on actual data, without having to acquire data from the NASA ASDC.

- A ZIP archive containing all intermediary, final and ancillary results obtained while generating the Figures of this manuscript has been assembled to document every step taken in this process.

- A User Manual (in PDF format) to explain how to acquire, install and use this second archive.

These five resources are available on the Research Data Repository of GFZ Data Services. We are currently waiting for the assignment of their final DOIs to synchronize the cross-references and update the PDF documents.

The diagram attached to this message shows how these various files and resources interact and complement each other: Reddish boxes refer to input data, the green box represents the software package, bluish boxes stand for the outputs generated by the software, and yellow boxes contain documents in PDF format. Except for the ESSD manuscript (available from this web site), and the 'Custom' input and output files, which will depend on the individual needs of each user, the other 5 files are available through the GFZ Data Services (pending DOI assignments).

Last but not least, the most current version of the manuscript will be provided as soon as we know the final DOIs assigned to each and every resource, so that all crossreferences can be synchronized.

Best regards, Michel Verstraete on behalf of all co-authors.
* * *
[Figure]

[Figure]

**Fig. 1.** Logical relations between the various files and resources generated to address the Editor comments.

---

## Author Response (AR1)

**Responses to the Reviewers of manuscript essd-2019-77**

- The comment of Catherine Moroney during the public discussion phase was a matter of clarification that was addressed in our response to that comment.

- The comment of the first Reviewer concerned the evaluation of the results which was addressed by inserting an entirely new section (5: Algorithm performance, pages 14 to 21 of the most current version of the manuscript).

- The comment of Editor Birgit Heim has been formally answered on the ESSD web site. As discussed extensively by email, 5 external resources have been made available to GFZ and are awaiting their DOIs, as follows:

1 A stand-alone, self-contained, executable version of the software to process MISR RCCM data and replace missing values, using the IDL Virtual Machine technology.

2 A User Manual (in PDF format) explaining how to acquire, install and use this software package on a PC running under Microsoft Windows operating system.

3 A ZIP archive of MISR L1B2 GRP and RCCM files for Path 168 and Orbit 68050 to serve as an example of input data: this allows users to explore how this software works on actual data, without having to acquire data from the NASA ASDC.

4 A ZIP archive containing all intermediary, final and ancillary results obtained while generating the Figures of this manuscript has been assembled to document every step taken in this process.

5 A User Manual (in PDF format) to explain how to acquire, install and use this second archive.

- When all DOIs will have been assigned, all relevant files (as well as the software source codes) will be updated to synchronize the cross-references and update the PDF documents. The final versions will then be made available to GFZ and ESSD as appropriate.

Please let us know if anything else should be provided at this point.

Best regards, Michel Verstraete on behalf of all co-authors.

---

## Author Response (AR2)

Here is a summary of how I have implemented the last round of requested changes for the manuscript essd-2019-77:

1. The abstract is changed as follows:

Original:

The Multi-angle Imaging SpectroRadiometer (\textsc{MISR}) is one of the five instruments hosted on-board the NASA Terra platform, launched on 18 December 1999. This instrument has been operational since 24 February 2000 and is still acquiring Earth Observation data as of this writing. The primary missions of \textsc{MISR} are to document the state and properties of the atmosphere, and in particular the clouds and aerosols it contains, as well as the planetary surface, on the basis of 36 data channels collectively gathered by its nine cameras (pointing in different directions along the orbital track) in four spectral bands (blue, green, red and near-infrared). The Radiometric Camera-by-Camera Cloud Mask (\textsc{RCCM}) is derived from the calibrated measurements at the nominal top of the atmosphere, and is provided separately for each of the nine cameras. This \textsc{RCCM} data product is permanently archived at the NASA Atmospheric Science Data Center (ASDC) in Langley, VA, USA and is openly accessible (\cite{Diner:1999:ATBD-RCCM} and \texttt{https://doi.org/10.5067/Terra/MISR/MIRCCM\_L2.004}). For various technical reasons described in this paper, this \textsc{RCCM} product exhibits missing data, even though an estimate of the clear or cloudy status of the environment at each individual observed location can be deduced from the available measurements. The aims of this paper are (1) to describe how to replace over 99\% of the missing values by estimates and (2) to briefly describe the software to replace missing \textsc{RCCM} values, which is openly available to the community from the GitHub web site \texttt{\url{https://github.com/mmverstraete}} or \texttt{https://doi.org/10.5281/zenodo.3240018}. Limited amounts of updated \textsc{MISR} \textsc{RCCM} data products are also archived in South Africa and can be made available upon request.

Updated:
* * *
The Multi-angle Imaging SpectroRadiometer (\textsc{MISR}) is one of the five instruments hosted on-board the NASA Terra platform, launched on 18 December 1999. This instrument has been operational since 24 February 2000 and is still acquiring Earth Observation data as of this writing. The primary missions of \textsc{MISR} are to document the state and properties of the atmosphere, and in particular the clouds and aerosols it contains, as well as the planetary surface, on the basis of 36 data channels collectively gathered by its nine cameras (pointing in different directions along the orbital track) in four spectral bands (blue, green, red and near-infrared). The Radiometric Camera-by-Camera Cloud Mask (\textsc{RCCM}) is derived from the calibrated measurements at the nominal top of the atmosphere, and is provided separately for each of the nine cameras. This \textsc{RCCM} data product is permanently archived at the NASA Atmospheric Science Data Center (ASDC) in Hampton, VA, USA and is openly accessible (\cite{Diner:1999:ATBD-RCCM} and \texttt{url{https://doi.org/10.5067/Terra/MISR/MIRCCM\_L2.004}}). For various technical

reasons described in this paper, this \textsc{RCCM} product exhibits missing data, even though an estimate of the clear or cloudy status of the environment at each individual observed location can be deduced from the available measurements. The aims of this paper are (1) to describe how to replace over 99\% of the missing values by estimates and (2) to briefly describe the software to replace missing \textsc{RCCM} values, which is openly available to the community from the GitHub web site \texttt{\url{https://github.com/mmverstraete}} or \texttt{https://doi.org/10.5281/zenodo.3240018}. Two additional sets of resources are also made available on the Research Data Repository of GFZ Data Services in conjunction with this paper: (A) The first set (\cite{Verstraete:2020:RCCM-Out}, \url{https://doi.org/10.5880/fidgeo.2020.004}) includes 3 items: (A1) a compressed archive \texttt{RCCM\_Out.zip} containing all intermediary, final and ancillary outputs created while generating the Figures of this manuscript, (A2) a User Manual \texttt{RCCM\_Out.pdf} describing how to install, uncompress and explore those files, and (A3) a separate input MISR data archive \texttt{RCCM\_input\_68050.zip} for Path 168, Orbit 68050. This latter archive is usable with (B) the second set (\cite{Verstraete:2020:RCCM-Soft-Win}, \url{https://doi.org/10.5880/fidgeo.2020.008}), which includes (B1) a stand-alone, self-contained, executable version of the RCCM correction codes \texttt{RCCM\_Soft\_Win.zip} using the IDL Virtual Machine technology that does not require a paid IDL license, as well as (B2) a User Manual \texttt{RCCM\_Soft\_Win.pdf} to explain how to install, uncompress and use this software.
* * *
2. Concerning the headline "Sample availability", and following our latest email exchanges, I have replaced the text in that section by a slightly edited version of the same information in the abstract:
* * *
Two sets of resources are made available on the Research Data Repository of GFZ Data Services in conjunction with this paper: (A) The first set (\cite{Verstraete:2020:RCCM-Out}, \url{https://doi.org/10.5880/fidgeo.2020.004}) includes 3 items: (A1) a compressed archive \texttt{RCCM\_Out.zip} containing all intermediary, final and ancillary outputs created while generating the Figures of this manuscript, (A2) a User Manual \texttt{RCCM\_Out.pdf} describing how to install, uncompress and explore those files, and (A3) a separate input MISR data archive \texttt{RCCM\_input\_68050.zip} for Path 168, Orbit 68050. This latter archive is usable with (B) the second set (\cite{Verstraete:2020:RCCM-Soft-Win}, \url{https://doi.org/10.5880/fidgeo.2020.008}), which includes (B1) a stand-alone, self-contained, executable version of the RCCM correction codes \texttt{RCCM\_Soft\_Win.zip} using the IDL Virtual Machine technology that does not require a paid IDL license, as well as (B2) a User Manual \texttt{RCCM\_Soft\_Win.pdf} to explain how to install, uncompress and use this software.
* * *
and moved this paragraph in the Section "Data availability". The earlier heading "Sample availability" has been deleted altogether.

3. I have also updated the manuscript on page 3 to note that MISR "offers a unique opportunity to study environmental issues over a continuous span of 20+ years.", instead of 19+ as was mentioned in the preprint submitted last year.

4. I have modified the LaTeX source code of the manuscript to use a "[$\bullet$]" symbol to denote items in all lists, rather than the default "-" symbol, because the latter could the mistaken for a minus sign on page 4. Using the same convention throughout the manuscript also seemed more coherent than using different conventions on different pages.

5. I have deleted the reference to the IDL routines to process AGP files on page 23 because this particular repository is not actually used in the RCCM case: it is required for the L1B2 paper, though.

6. The incorrect references to the GFZ DOIs were due to the insertion of the full https address in the DOI fields of the BibTeX file, while the double hyperlink associated with the citation to the paper by Diner et al. was due to an extra blank space: both problems have been addressed.

7. Please remember that while the DOIs of the GFZ resources are known, the definitive DOI of the main manuscript is still unknown, as of this writing: I will thus have to update the two User Manuals, to point to the final version of the manuscript, rather than the preprint 'Discussion' paper. That is not a major problem because we can upgrade the content of the GFZ web pages without changing their DOIs.

8. However, when I will update the IDL software routines on my GitHub page to point to the citation and DOI of the final manuscript, Zenodo will automatically assign new DOIs to those software repositories: hence, I will need to provide you with those updated references after the ESSD manuscript DOI is finalized. You may want to warn the staff in charge of the final publication that this extra step should be expected.

Lastly, the PDF version of the manuscript, with those changes implemented, is attached below (file 'MISR_RCCM_v13_small3.pdf'). I have used the commercial tool "PDF Expert" to reduce the size of the PDF file generated by LaTeX from 13.0 to 2.9 MB, with no apparent effect on my MacBook Pro. However, if that causes production problems, I'll be able to provide you with the larger version: Let me know if you need this.